# Vascular Diameter as Clue for the Diagnosis of Clinically and/or Dermoscopically Equivocal Pigmented and Non-Pigmented Basal Cell Carcinomas and Nodular Melanomas

**DOI:** 10.3390/medicina58121761

**Published:** 2022-11-30

**Authors:** Roberta Giuffrida, Claudio Conforti, Andreas Blum, Marija Buljan, Fabrizio Guarneri, Rainer Hofmann-Wellenhof, Caterina Longo, John Paoli, Cliff Rosendahl, H. Peter Soyer, Ružica Jurakić Tončić, Roberta Vezzoni, Iris Zalaudek

**Affiliations:** 1Department of Clinical and Experimental Medicine, Dermatology, University of Messina, 98124 Messina, Italy; 2Department of Dermatology and Venereology, Dermatology Clinic, Maggiore Hospital, University of Trieste, 34125 Trieste, Italy; 3Hautarzt-und Lehrpraxis Konstanz, Augustinerplatz 7, 78462 Konstanz, Germany; 4Department of Dermatology and Venereology, Sestre Milosrdnice University Hospital Centre, 10000 Zagreb, Croatia; 5School of Dental Medicine, University of Zagreb, 10000 Zagreb, Croatia; 6Nonmelanoma Skin Cancer Unit, Department of Dermatology and Venerology, Medical University of Graz, 8036 Graz, Austria; 7Department of Dermatology, University of Modena and Reggio Emilia, 41124 Modena, Italy; 8Centro Oncologico ad Alta Tecnologia Diagnostica, Azienda Unità Sanitaria Locale-IRCCS di Reggio Emilia, 42123 Reggio Emilia, Italy; 9Department of Dermatology, Institute of Clinical Sciences, Sahlgrenska Academy, University of Gothenburg, 40530 Gothenburg, Sweden; 10School of Medicine, The University of Queensland, Brisbane, QLD 4102, Australia; 11Dermatology Research Centre, The University of Queensland Diamantina Institute, The University of Queensland, Brisbane, QLD 4102, Australia; 12Dermoscopy Unit, University of Zagreb, 10000 Zagreb, Croatia

**Keywords:** skin cancer, vessels, dermoscopy, vascular diameter, melanoma, basal cell carcinoma

## Abstract

*Background and objectives*: Dermoscopy is a useful tool for the early and non-invasive diagnosis of skin malignancies. Besides many progresses, heavily pigmented and amelanotic skin tumors remain still a challenge. We aimed to investigate by dermoscopy if distinctive morphologic characteristics of vessels may help the diagnosis of equivocal nodular lesions. *Materials and Methods:* A collage of 16 challenging clinical and dermoscopic images of 8 amelanotic and 8 heavily pigmented nodular melanomas and basal cell carcinomas was sent via e-mail to 8 expert dermoscopists. *Results:* Dermoscopy improved diagnostic accuracy in 40 cases. Vessels were considered the best clue in 71 cases. Focusing on the diameter of vessels improved diagnosis in 5 cases. *Conclusions:* vascular diameter in addition to morphology and arrangement may be a useful dermoscopic clue for the differential diagnosis of clinically equivocal nodular malignant tumors.

## 1. Introduction

Several lines of evidence demonstrate the value of dermoscopy in the diagnosis of pigmented and non-pigmented skin tumors [1,2]. Besides much progress being made, heavily pigmented and amelanotic skin tumors still remain a challenge. Although the so-called blue and black rule, defined as the coexistence of blue and black in a given nodular tumor, has been shown to accurately differentiate nodular melanoma (nMEL) from blue nevus, it may fail to accurately differentiate highly aggressive nMEL from a low-risk heavily pigmented nodular basal cell carcinoma (nBCC) [3]. This is related to the histopathological correlates of blue and black, with blue corresponding to dermal melanin and black to melanin or hemoglobin (in areas of ulceration) in the upper epidermal layers. In the realm of amelanotic nodules, arborizing vessels defined as large caliber branching vessels represent an important diagnostic criterion of nBCC, whereas polymorphous vessels, defined as vessels with more than one morphology, are considered a clue for any kind of malignancy [4]. For both, nBCC and nMEL, surgery is considered the first-line treatment, but with different urgency. Improving the differential diagnosis between equivocal nBCC and nMEL may, therefore, have a practical impact in assessing the priority for surgery.

We recently noticed by reviewing our database that heavily pigmented BCCs commonly exhibit blue-to-blue-black structures in addition to some large caliber vessels, whereas such vessels were never seen in heavily pigmented nMELs.

The latter observation may be explained by another feature we noticed, namely, that amelanotic nMEL in contrast to nBCC or other types of low-risk nodular carcinomas (i.e., keratoacanthoma) typically reveal dense, short and fine, polymorphic small caliber micro-vessels, which may be easily covered by the presence of heavy pigmentation in the case of nMEL. In contrast, even large amounts of pigmentation do not completely hide the classical large superficial macro-vessels of nBCCs.

## 2. Methods

We tested our hypothesis by sending a collage of 16 challenging clinical images of 8 amelanotic and 8 heavily pigmented nodular lesions (8 melanomas and 8 BCCs) and their corresponding dermoscopic images (examples in Figure 1) to 8 expert dermoscopists (128 evaluations in total) via e-mail. 

In a first round, the dermatologists were asked to make a diagnosis only on the clinical images, then on the dermoscopic images, explaining the dermoscopic clues useful for the diagnosis. Finally, they were asked to confirm or change their diagnoses, focusing on the diameter of vessels visible in the lesions, as clue for malignancy.

## 3. Results

Dermoscopy improved diagnostic accuracy in 40 cases. Vessels were considered the best clue in 71 cases, with arborizing vessels as most prevalent clue of BCCs (35/64 observations). Only in 3 cases, a diagnosis of melanoma was associated with the observation of small diameter vessels as best dermoscopic findings. Prominent large caliber vessels were correctly considered best clue for the diagnosis of BCC in 4 cases. Focusing on the diameter of vessels improved diagnosis in 5 cases.

The results of our survey are shown in Table 1.

Dermoscopic vessels features considered best diagnostic clue are listed in Table 2.

## 4. Discussion

The use of dermoscopy in clinical practice has been shown to increase diagnostic accuracy both for pigmented and nonpigmented skin tumors [5,6].

While specific dermoscopic features are easily recognizable in most pigmented tumors, when assessing nonpigmented or hypopigmented skin tumors, reaching out for a diagnosis can be a challenge and the visualization of vessels, usually not visible to the naked eye, with the identification of a characteristic morphology can be the only key to the dermoscopic diagnosis.

Moreover, regardless of pigmentation, the dermoscopic recognition of nodular malignant skin lesions is also difficult, due to the fact that the tumor often lacks the specific criteria for malignancy [6,7].

In 2001, the Board of the Consensus Net Meeting on Dermoscopy discussed a two-step algorithm for the classification of pigmented skin lesions [8].

In 2010, a revised two-step algorithm has been described, taking into consideration polarized dermoscopy and vascular architecture [9].

In the same year, a three-step algorithm for dermoscopic diagnosis of non-/hypopigmented skin tumors, including vascular morphology, architectural arrangement of vessels (i.e., regular, in a string, clustered, radial, irregularly branched and irregular) and other optional criteria (such as a halo surrounding the structures, residual pigmentation, hairs, central duct openings, superficial scales, ulceration, etc.) has been proposed [10]. Among these, the morphology of vessels has been considered to be of primary importance.

To date, six morphologic categories of vascular patterns have been described in literature, including comma-like vessels, dotted vessels, linear irregular vessels, hairpin vessels, glomerular vessels and arborizing vessels [10]. Except for certain cases, comma, dotted and linear-irregular vessels are indicative of melanocytic lesions, whereas hairpin, glomerular and arborizing vessels are present in keratinocytic tumors [10]. The presence of vessels of mixed morphology in the same lesion (polymorphous vessels) is considered to be suspicious of malignant skin tumors [11].

Moreover, it is necessary to remember that clinically flat and superficial tumors usually show different vascular morphology compared with their thick and nodular counterparts [10].

Sometimes highly aggressive nMEL may be difficult to distinguish from low-risk nodular keratinocyte cancer, such as nBCC.

Surgical excision is the first-choice treatment for both entities. Nodular basal cell carcinoma usually doesn’t need an urgent referral, whereas in case of nMEL, early recognition and surgical removal is key for maximizing survival outcomes of patients.

Based on our personal clinical experience and according to our preliminary results, the observation of vessels caliber and density by dermoscopy may help in distinguishing tumors with a longstanding progressive and indolent growth (i.e., nBCCs) from tumors with aggressive behavior (i.e., nMELs).

When dermoscopy of nodular non-/hypopigmented lesion reveals small caliber vessels, usually with high density, a suspicion of highly aggressive malignancy should be considered in order to perform a biopsy and obtain a prompt histological diagnosis. Indeed, we found that non-/hypopigmented nodular melanoma typically shows densely packed vessels of small diameter (Figure 2), whereas, for example, nodular non-/hypopigmented basal cell carcinoma is characterized by arborizing vessels of larger caliber with a low density (Figure 3).

A similar situation can be observed confronting nodular poorly differentiated squamous cell carcinoma (SCC) with keratoachantoma, that are typically raised non pigmented skin lesions. Hairpin-like vessels occur more frequently in well differentiated SCC (such as keratoacanthoma) [10]. They usually appear elongated and thicker, distributed around the periphery of the lesion. Vessels in invasive SCC are typically polymorphic/non-homogenous with small caliber [12].

In case of pigmented nMEL, structureless blue pigmentation, areas of pink and black colors and/or ulceration can be the clues for the diagnosis. Vascular structures, when visible, are located at the periphery of the lesions as polymorphous vessels and milky-red globules [13] (Figure 4).

Pigmented nodular BCCs are instead characterized on dermoscopy by blue-grey ovoid nests or blue-grey dots/globules usually associated with arborizing vessels (Figure 5). At the periphery of the tumor structures corresponding to dermo-epidermal pigmentation (maple leaf-like areas, spoke wheel areas and/or concentric structures) can be found [14].

In both cases, based on our personal experience, in the absence of any specific dermoscopic features, an aggressive behavior (i.e., nMEL) should be hypothesized when on dermoscopy the lesion exhibits higher density of vessels of small diameter. Conversely, an indolent course (i.e., nBCC) should be assumed for tumors with lower density of larger caliber vessels, typically visible also in presence of heavy pigmentation.

These considerations could be useful in the triage setting of equivocal and suspicious hypo-/hyperpigmented nodular lesions, in order to identify those requiring prompt biopsy.

## 5. Conclusions

These preliminary results suggest that vascular diameter in addition to morphology and arrangement may be a useful dermoscopic clue for the differential diagnosis of clinically equivocal nodular malignant tumors.

Based on our experience, dense and small caliber vessels in nodular lesions are suspicious for high malignancy (although sometimes hidden by heavy pigmentation in nMEL), whereas larger superficial vessels with low density suggest a locally aggressive behavior.

These preliminary observations on the role of vascular diameter as a new diagnostic dermoscopic clue of malignancy should be validated in a prospective study, collecting a large series of nodular lesions. The differentiation of clinically and/or dermoscopically equivocal pigmented and non-/hypopigmented nodular BCCs and melanomas is indeed important especially in order to expedite surgical treatment of aggressive tumors and provide the best care, as early as possible.

## Figures and Tables

**Figure 1 medicina-58-01761-f001:**
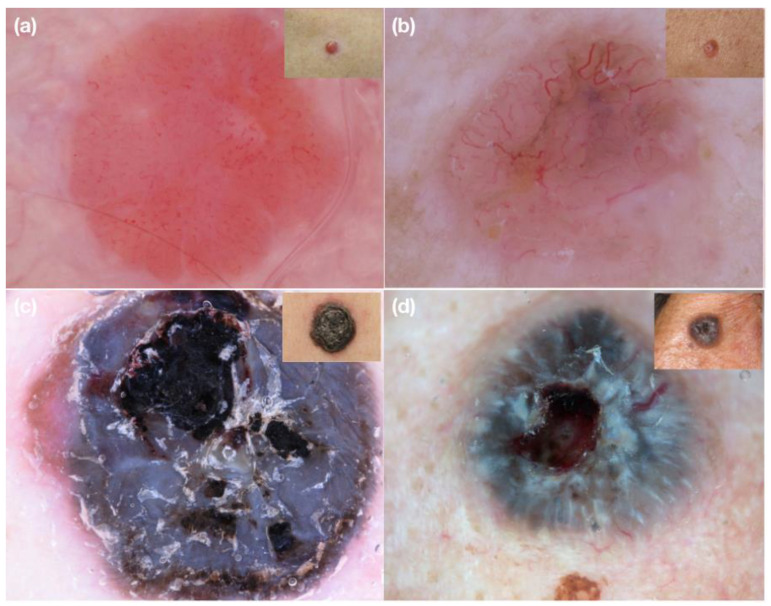
Examples of clinical and dermoscopic images concerning challenging nodular lesions submitted for evaluation to the expert dermoscopists involved in the study. (**a**) Nodular amelanotic melanoma; (**b**) nodular non-pigmented basal cell carcinoma; (**c**) nodular heavily pigmented melanoma; (**d**) nodular pigmented basal cell carcinoma.

**Figure 2 medicina-58-01761-f002:**
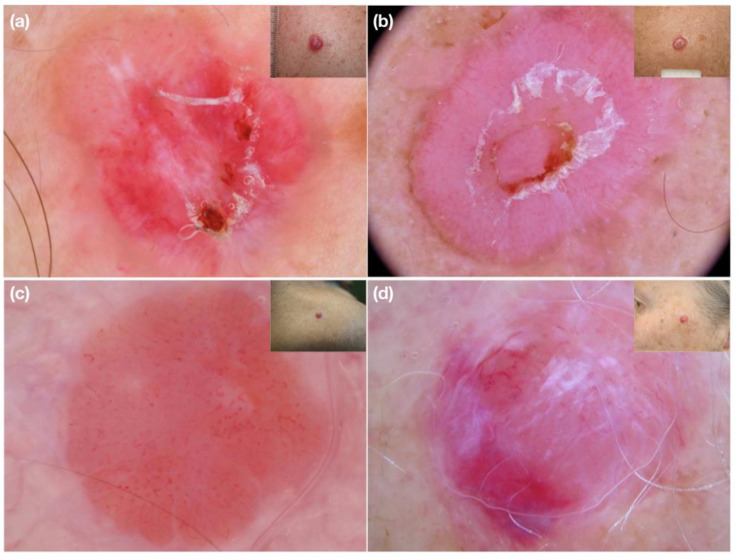
(**a**–**d**) Clinical and dermoscopic images of nodular amelanotic melanomas. Dermoscopy shows dense, short and fine polymorphic micro-vessels.

**Figure 3 medicina-58-01761-f003:**
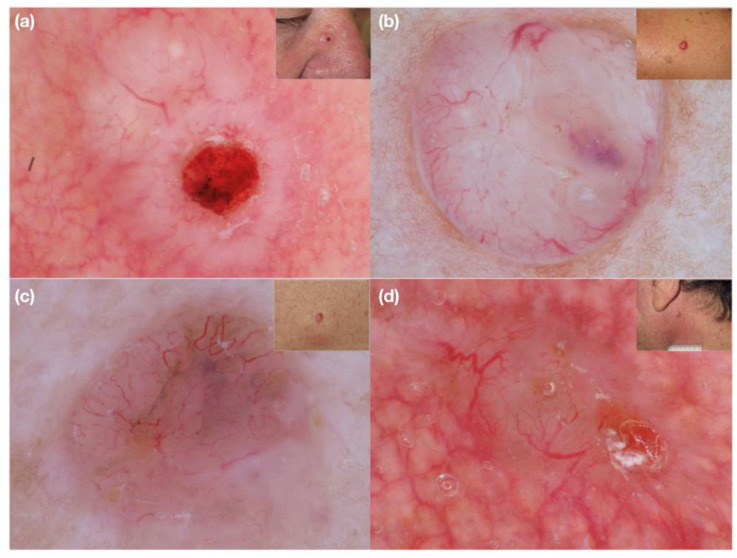
(**a**–**d**) Clinical and dermoscopic images of nodular non-pigmented basal cell carcinoma. Large superficial macro-vessels are easily visible on dermoscopy.

**Figure 4 medicina-58-01761-f004:**
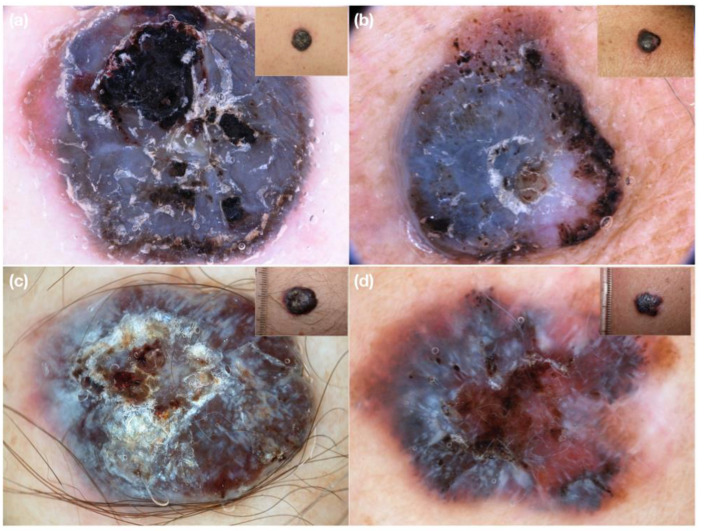
Clinical and dermoscopic images of nodular heavily pigmented melanoma. On dermoscopy, short polymorphous vessels can be seen at the periphery (**a**,**b**) or are hidden by heavy pigmentation (**c**,**d**).

**Figure 5 medicina-58-01761-f005:**
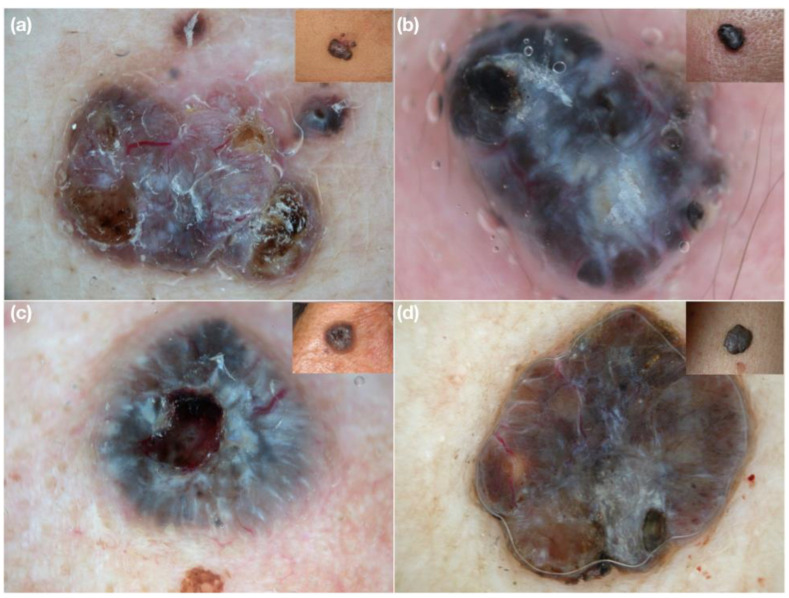
(**a**–**d**) Clinical and dermoscopic images of nodular pigmented basal cell carcinoma. The heavy pigmentation does not completely hide the superficial large caliber vessels of BCC on dermoscopy.

**Table 1 medicina-58-01761-t001:** Results of clinical and dermoscopic evaluation of 16 study images (concerning 8 nodular melanomas and 8 nodular basal cell carcinomas) by the 8 expert dermoscopists involved in the survey (128 evaluations in total).

	Number of Correct Diagnoses	% Improvement vs. Clinical Diagnosis
Clinical	65/128 (50.8%)	
Dermoscopic	105/128 (82.0%)	61.5%
Focusing on the diameter of vessels	110/128 (85.9%)	69.2%

**Table 2 medicina-58-01761-t002:** Dermoscopic vessels features considered best diagnostic clue in the evaluation of 16 study images (concerning 8 nodular melanomas and 8 nodular basal cell carcinomas) by the 8 expert dermoscopists involved in the survey (128 evaluations in total).

	Dermoscopic Evaluations	Nodular Melanoma		Nodular BCC	
	*n* = 71/128 (55.5%)	Non-Pigmented	Pigmented	Non-Pigmented	Pigmented
Arborizing vessels, *n* (%)	35 (27.3%)	1 (0.8%)	-	25 (19.5%)	9 (7.0%)
Comma vessels, *n* (%)	1 (0.8%)	-	-	1 (0.8%)	-
Linear irregular vessels, *n* (%)	7 (5.46%)	4 (3.1%)	-	2 (1.5%)	1 (0.8%)
Dotted vessels, *n* (%)	1 (0.8%)	1 (0.8%)	-	-	-
Curved vessels, *n* (%)	1 (0.8%)	1 (0.8%)	-	-	-
Hairpin-like vessels, *n* (%)	1 (0.8%)	1 (0.8%)	-	-	-
Corkcrew vessels, *n* (%)	1 (0.8%)	1 (0.8%)	-	-	-
Polymorphous vessels, *n* (%)	17 (13.2%)	14 (10.9%)		3 (2.3%)	-
Small vessels, *n* (%)	3 (2.3%)	3 (2.3%)	-	-	-
Large prominent vessels, *n* (%)	4 (3.1%)	-	-	-	4 (3.1%)

BCC = basal cell carcinoma.

## Data Availability

Not applicable.

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
