# Peer review of "Vascular Diameter as Clue for the Diagnosis of Clinically and/or Dermoscopically Equivocal Pigmented and Non-Pigmented Basal Cell Carcinomas and Nodular Melanomas"

_medicina, 2022, doi:10.3390/medicina58121761_

Round 1

Reviewer 1 Report

The present paper gives a reasonable and useful message for dermatologists' clinical diagnostic activity, therfore I recommend publication.

I enclose few suggestions that can imporve the overall quality:

1- figure 1d. in this image of a nodular pigmented basal cell carcinoma we cannot see (at least at this enlargement) the large caliber vessels. I suggest to replace this dermoscopoic picture with another case, to support the statement ".....heavily pigmented BCCs commonly exhibit blue to blue-black structures in addition to some large caliber vessels".

-Table 1. Results of clinical and dermoscopic evaluation of 16 study images (concerning 8 nodular 68 melanomas and 8 nodular basal cell carcinomas) by the 8 expert dermoscopists involved in the survey (128 evaluations in total).it could be possible to include in this table the average rates of assessement of the differnet vessels features? (ie., arborizing, small-diamter, short-fine..etc..) since in tyhe text is stated "Vessels were considered the 74 best clue in 71 cases, with arborizing vessels as most prevalent clue of BCCs (35/64 observations). Only in two cases, a diagnosis of melanoma was associated with the observation of small diameter vessels as best dermoscopic findings.Focusing on the diameter of vessels improved diagnosis in 5 cases "

Author Response

Dear Reviewer,

Thank you for taking time for the revision of our manuscript and for your nice comments.

Please find enclosed herewith the revised version of our manuscript “Vascular diameter as clue for the diagnosis of clinically and/or dermoscopically equivocal pigmented and non-pigmented basal cell carcinomas and nodular melanomas” (Manuscript ID medicina-2008667_R1).

Please, also find below  a detailed list of our point-to-point answers to the comments received and modifications made accordingly (highlighted in yellow in the text).

Thank you for your suggestions.

Best regards

Roberta Giuffrida, MD, PhD

Department of Clinical and Experimental Medicine,

Dermatology, University of Messina, Messina, Italy

- ANSWERS TO REVIEWER'S COMMENTS -

1) As suggested, we replaced the BCC reported in figure 1d (and accordingly in figure 5c) with another BCC with pigmentation and vessels better visible.

2) We added a new table (table 2) listing all the vascular features observed considering vessels as best dermoscopic clue. A sentence about the presence of large prominent caliber vessels clue of BCC was also added. Thanks for your suggestion.

Reviewer 2 Report

Thanks for bringing this observation to the attention of dermatologists. Experienced dermoscopists will confirm the usefulness of vascular morphology in the triage of nonpigmented and deeply pigmented nodules.  

Please correct a numbering format error for the first three references.

Author Response

Dear Reviewer,

Thank you for taking time for the revision of our manuscript and for your positive comments.

Please find enclosed herewith the revised version of our manuscript “Vascular diameter as clue for the diagnosis of clinically and/or dermoscopically equivocal pigmented and non-pigmented basal cell carcinomas and nodular melanomas” (Manuscript ID medicina-2008667_R1).

Please, also find below  our  answer to the comment received and modifications made accordingly (highlighted in yellow in the text).

Thank you for your suggestions.

Best regards

Roberta Giuffrida, MD, PhD

Department of Clinical and Experimental Medicine,

Dermatology, University of Messina, Messina, Italy

- ANSWERS TO REVIEWER'S COMMENTS -

We revised all the references and made corrections according to authors’ guideline. Thank you.
